# Possibilities for Utilization of Cherry Products (Juice and Pomace) in Beer Production

Petar Nedyalkov [1], Ivan Bakardzhiyski [1] , Vasil Shikov [2], Maria Kaneva [1] and Vesela Shopska [1,*]

[1] Department of Wine and Beer Technology, University of Food Technologies—Plovdiv, 26 Maritsa Boulevard, 4002 Plovdiv, Bulgaria; p_nedyalkov@uft-plovdiv.bg (P.N.); ivanbak81@yahoo.com (I.B.); m_kaneva@uft-plovdiv.bg (M.K.)

[2] Department of Food Preservation and Refrigeration Technology, University of Food Technologies—Plovdiv, 26 Maritsa Boulevard, 4002 Plovdiv, Bulgaria; vshikov@uft-plovdiv.bg

* Correspondence: vesi_nevelinova@abv.bg; Tel.: +359-887263065

**Abstract:** Fruit addition can enrich beer with flavor and bioactive substances. Sweet cherry (*Prunus avium* L.) can be added in beer as a whole fruit, fruit juice, or pulp, but there is no data for the addition of cherry pomace in beer. Therefore, we investigated the addition of cherry juice and pomace during beer fermentation on the first and seventh day and studied the basic beer parameters (alcohol and extract), sensorial evaluation, phenolic compounds, and antioxidant activity of the beers produced, measured using six different methods (DPPH, ABTS, FRAP, CUPRAC, ORAC, and HORAC) and compared the results with a control sample without cherry products addition. The results showed a strong correlation between the antioxidant activity values obtained using the DPPH, FRAP, CUPRAC, and HORAC methods and the concentration of phenolic compounds in the studied beers. The phenolic compound content and antioxidant activity increased when cherries juice or pomace were added. The increase was much more significant when pomace was used. Therefore, it can be concluded that cherry pomace addition is a better option than cherry juice for beer production because of the increased content of bioactive compounds and the sustainability of the beers obtained.

**Keywords:** beer; cherry juice; cherry pomace; phenolic compounds; antioxidant activity; sensory characteristics

## 1. Introduction

Beer is one of the oldest and most consumed beverages in the world. Beer contains four main ingredients, malt, water, hop, and yeast, but also additional ingredients such as fruits, herbs, and spices, or alternative fermentable substrates that can be used in beer production [1,2]. Such a wide range of raw materials that can be used in the production of beers and beer-based products provides a great opportunity for brewers to conquer new markets and to meet demands of unconventional consumer groups. In recent years, the market for special beers with improved health functions and/or with a new refreshing taste has significantly increased [3]. Therefore, fruit beers are becoming very popular all over the world. Fruit beers are produced by adding fruits, extracts, or flavorings. Fruits do not just impart new flavors, but they may also increase the concentration of bioactive components since they are extracted during the secondary fermentation and maturation of beers [4,5].

The addition of sour cherries (*Prunus cerasus* L.) is practiced traditionally in Belgium for the production of sour cherry lambic beer ('kriek), but the application of sweet cherry (*Prunus avium* L.) in beer production as a puree or fruit juice was also reported [2,5–8]. Sweet cherries contain approximately 13 g/100 g of simple sugars (glucose and fructose). They contain both hydrosoluble (C, B) and liposoluble (A, E and K) vitamins, fibers, some carotenoids (in particular beta-carotene, and to a lower extent lutein and zeaxantine), and some minerals such as calcium, magnesium, phosphorous, and potassium. Sweet

cherries also contain some phenolic compounds such as phenolic acids, flavonoids, and anthocyanins [9,10].

In fruit beer production, fruits can be added as a whole fruit, fruit juice, or pulp [11], but there is no data for the production of fruit beer with the addition of pomace (by-products from fruit juice production). Therefore, the aim of this study is to investigate the effect of the addition of sweet cherry juice and pomace at the same concentrations at different stages of fermentation on the physicochemical (alcohol, extract, proteins, total phenolic compounds, phenolic acids, and flavonoids), antioxidant, and sensorial properties of beer. The beers produced were compared to a control beer without cherry products.

## 2. Materials and Methods

### 2.1. Raw Materials

Pilsner malt (Weyermann, Bamberg, Germany), bitter Perle, and aromatic Cascade hops (Bulhops, Velingrad, Bulgaria) were used for wort production. Beer fermentation was carried out with dry yeast *Saccharomyces pastorianus* Saflager W 34/70 (Fermentis, Marquette-lez-Lille, France). The sweet cherries (*Prunus avium* L.) without stones were purchased in a frozen state from Bulfruct Ltd., Kostenets, Bulgaria, and kept at $-18\ ^\circ$C. Fruits were defrosted immediately before use and pressed through the Omega Sana Juicer EUJ-707R (EUJuicers S. A Sana, Seoul, Republic of Korea). The cherry juice (TSS = 18.1 °Brix) and pomace (TS = 20.76% ($w/w$)) obtained after pressing were used in beer fermentation.

### 2.2. Brewing

Wort with extract of $14 \pm 0.5$ °P was produced by mixing 15 kg coarsely ground Pilsen malt and 55 L water using Home Brew 50 (TM INOX, Plovdiv, Bulgaria). Mashing was conducted by increasing the temperature by 1 °C/min and maintaining rests at the following temperatures: 20 min at 60 °C, 20 min at 65 °C, 25 min at 72 °C, and 1 min at 78 °C. Lautering, sparging, and boiling were carried out in the same Home Brew 50. A total of 20 g of bitter hop was added 10 min after wort boiling, and 17.14 g of the aromatic hop was added 7 min before the end of the boiling in order to obtain wort with a total amount of $\alpha$-bitter acids of 60 mg/L. The hot trub was removed, and the wort was cooled to the fermentation temperature.

### 2.3. Beer Fermentation

The wort was aerated and transferred into a stainless steel cylindroconical fermenter (TM INOX, Plovdiv, Bulgaria) with a working volume of 50 L. The yeast was rehydrated according to the manufacturer's instructions and pitched into the wort. The pitched wort was transferred into smaller stainless-steel fermenters (TM INOX, Plovdiv, Bulgaria) with a working volume of 4 L, where the cherry juice and pomace were added according to Table 1.

**Table 1.** Scheme of addition of cherry products during fermentation.

| No. | Variant | Addition of Cherry Products | Concentration | Day of Fermentation |
|---|---|---|---|---|
| 1 | Control | - | - | - |
| 2 | CHJ (1) | Cherry juice | 15% ($w/v$) | 1 |
| 3 | CHJ (7) | | | 7 |
| 4 | CHP (1) | Cherry pomace | 15% ($w/v$) | 1 |
| 5 | CHP (7) | | | 7 |

CHJ (1)—cherry juice addition on the first day of fermentation; CHJ (7)—cherry juice addition on the seventh day of fermentation; CHP (1)—cherry pomace addition on the first day of fermentation; CHP (7)—cherry pomace addition on the seventh day of fermentation.

The fermentation was carried out at 14 °C. The temperature was controlled by placing the fermenter in a refrigerator with a temperature controller. The main fermentation was monitored hydrometrically, and it continued until the difference between the attenuation

limit and apparent attenuation became approximately 20%. Beer maturation continued for 14 days at 14 °C, and beer lagering was carried out for 5 days at 2 °C. Both were conducted under pressure. All variants were carried out in duplicate. After lagering, the beer was bottled using a "beer gun" (Blichmann Engineering, Lafayette, IN, USA).

*2.4. Beer Analysis*

2.4.1. Sample Preparation

The beers were filtered through Macherey-Nagel MN 619 $\frac{1}{4}$ Ø 320 filter paper on the day of bottling and frozen. Before analysis, the beers were defrosted, parts of them were diluted with methanol in a proper ratio and left for 30 min at room temperature. After filtration with Whattman No.1 filter paper, they were used for the analysis of phenolic compounds content and antioxidant activity of beers.

2.4.2. Basic Beer Parameters

EBC standard methods were used for the determination of real extract (method 9.4) and alcohol (method 9.2.1) in beers [12].

2.4.3. Proteins Determination

Proteins were determined quantitatively as described in Nedyalkov et al. [13]. The concentration of protein was obtained using the Bradford method (1976), with Stosheck's modification (cited in [14]). The results were expressed in mg/L of bovine serum albumin (BSA).

2.4.4. Total Phenolic Compounds (TPC) Determination with Folin–Ciocalteu Reagent (FCR)

TPC with FC reagent were determined as described in Shopska et al. [15]. A total of 1 mL of methanol extract was mixed with 4 mL of freshly prepared FCR (1:10 $v/v$ with water). Then, 5 mL of sodium carbonate (7.5%, $w/v$) was added to the mixture. After incubation at room temperature for 60 min, the absorbance of the mixture was read at 765 nm against a blank sample prepared with distilled water. The results were expressed as mg of gallic acid equivalents (GAE)/L.

2.4.5. Phenolic Compounds Determination Using Modified Glories Methods

The content of phenolic acids and flavonoids was determined using a modified Glories method [16]. A mixture of 1 mL of 0.1% HCl in 95% ethanol ($v/v$), 18.2 mL of 2% HCl ($v/v$), and 1 mL of methanol extract was kept for 15 min at room temperature. A blank prepared with distilled water was used for the determination of phenolic acids (PAs) at 320 and flavonoids (F) at 360 nm. Calibration curves were used for results' expression as mg caffeic acid equivalent/L (CAE/L) for PAs and mg quercetin equivalent/L (QE/L) for F, respectively.

2.4.6. Antioxidant Activity (AOA) of Beers

- DPPH(2,2′-Diphenyl-1-picrylhydrazyl) radical scavenging activity

AOA was measured using a modification of the DPPH method as described in Shopska et al. [15]. A total of 0.25 mL of methanol extract was added to 2.25 mL of 0.06 mM methanol solution of DPPH. The absorbance at 517 nm against a blank prepared with distilled water was read after being left for 30 min in the dark. The percentage inhibition was calculated against a control, prepared with methanol, and compared to a Trolox standard curve. The results were expressed in μmol Trolox Equivalent (TE)/L.

- FRAP (Ferric Reducing Ability of Plasma) Method

The FRAP analysis was performed according to the method described by Benzie and Strain [17], with some modifications. Briefly, the FRAP reagent was prepared by mixing 300 mM sodium acetate and glacial acetic acid buffer (pH 3.6), 20 mM ferric chloride hexahydrate, and 10 mM 4,6-tripryridyl-s-triazine (TPTZ) made up in 40 mM HCl in the

ratio of 10:1:1. The FRAP assay was performed by adding 0.15 mL of sample to 2.85 mL of reagent and incubating for 4 min in the dark. Readings were taken at 593 nm against a blank prepared with methanol. The antioxidant activity was determined using a calibration curve made with Trolox. The results were expressed in μmol TE/L.

- ABTS (2,2′-Azinobis-(3-ethylbenzothiazoline-6-sulfonate)) radical cation scavenging activity

The beer's ability to scavenge the ABTS$^+$ free radical was investigated using a modified methodology previously reported by Iqbel et al. [18]. Equal amounts of 2.45 mM potassium persulfate and 7 mM ABTS were reacted for 12–16 h in the dark. The solution was then diluted with methanol in a ratio of 1:30 to an absorbance of $1.1 \pm 0.1$ at 734 nm to form the test reagent. Reaction mixtures containing 0.15 mL of sample and 2.85 mL of reagent were incubated in the dark for 30 min. The absorbance was taken at 734 nm against methanol. The percentage inhibition was calculated against a control sample, prepared with methanol, and compared to a Trolox standard curve. The results were expressed in μmol TE/L.

- CUPRAC (Cupric Reducing Antioxidant Capacity) Method

The CUPRAC assay was performed as described in Apak et al. [19]. A total of 0.5 mL of the methanol extract was mixed with 1 mL each of ammonium acetate buffer (pH 7), 7.5 mM neocuproine in ethanol, and 10 mM copper (II) chloride dihydrate solution. At the end, 0.6 mL of distilled water was added and the mixture was left for 30 min at room temperature. The absorbance was recorded at 450 nm against a blank prepared with distilled water. The standard curve was established using Trolox. The results were expressed in μmol TE/L.

- ORAC (Oxygen Radical Absorbance Capacity) Method

The ORAC analysis was carried out according to Denev et al. [20], with some modifications as described in Shopska et al. [15]. Analyses were conducted in 75 mM phosphate buffer pH 7.4. A total of 170 μL of 70 nM fluorescein and 10 μL of the sample were incubated for 20 min at 37 °C directly in the FLUOstar OPTIMA fluorimeter (BMG LABTECH, Ortenberg, Germany). A peroxyl radical was generated using 20 μL of 51.5 mM 2, 2′-azobis (2-amidino-propane) dihydrochloride, which was prepared fresh for each run. Fluorescence conditions were as follows: excitation at 485 nm and emission at 520 nm. The mixture was automatically shaken, and the fluorescence was read every minute until reaching a zero value. To express the AOA, a standard curve with Trolox solutions was used. The results were expressed in μM TE/L.

- HORAC (Hydroxyl Radical Antioxidant Capacity) Method

The HORAC analysis was carried out as described in Denev et al. [20]. A total of 170 μL (60 nM, final concentration) fluorescein and 10 μL of sample were incubated at 37 °C for 20 min directly in the FLUOstar OPTIMA fluorimeter (BMG LABTECH, Ortenberg, Germany). After the addition of 10 μL $H_2O_2$ (27.5 mM, final concentration) and 10 μL of Co(II) (230 μM, final concentration) solutions, the initial fluorescence was measured against a blank prepared with phosphate buffer. Fluorescence conditions were as follows: excitation at 485 nm and emission at 520 nm. The mixture was automatically shaken, and the fluorescence was read every minute until reaching a zero value. To express the AOA, a standard curve with gallic acid solutions was used. The results were expressed in μGAE TE/L.

### 2.4.7. Sensory Analysis

Sensory evaluation was conducted 14 days after beer bottling by a trained 5-member tasting panel, using the descriptive method (method 13.10) and the ranking test (method 13.11) of EBC [12].

### 2.5. *Statistical Analysis*

The results are given as mean value ± standard deviation of 3 experiments. One-way ANOVA and Scheffe's multiple range test as described by Donchev et al. [21] at $p < 0.05$

were used to identify and specify the significant differences. All the statistical analyses were carried out using Microsoft Excel[TM] functions.

## 3. Results

### 3.1. Basic Characteristics—Alcohol Content and Real Extract

The results for basic beer characteristics (alcohol content and real extract) are presented in Table 2. The cherry products' addition led to an increase in the beer alcohol content with approximately 0.3–0.4% (*v/v*). It can be ascribed to the sugar content of cherry juice and pomace, which was converted to alcohol during the alcohol fermentation. However, there was no significant difference between the alcohol content of beers produced with cheery products. Moreover, the time of addition of cherry products did not affect the alcohol content of the beer. In terms of real extract, beers with cherry product addition did not differ significantly from the control beer. Therefore, it can be concluded that all the fermentable sugars from wort and cherry products had been utilized by the yeast during fermentation.

**Table 2.** Alcohol content and real extract of the beer samples.

| Beer Sample | Alcohol Content, % (*v/v*) | Real Extract, % (*w/w*) |
|---|---|---|
| Control | 6.07 ± 0.07 | 4.62 ± 0.01 [b] |
| CHJ (1) | 6.40 ± 0.06 [a] | 4.52 ± 0.12 [b] |
| CHJ (7) | 6.38 ± 0.13 [a] | 4.57 ± 0.06 [b] |
| CHP (1) | 6.45 ± 0.08 [a] | 4.50 ± 0.11 [b] |
| CHP (7) | 6.39 ± 0.09 [a] | 4.65 ± 0.07 [b] |

The same letter for a given sample means no significant differences (95% confidence level). CHJ (1)—cherry juice addition on the first day of fermentation; CHJ (7)—cherry juice addition on the seventh day of fermentation; CHP (1)—cherry pomace addition on the first day of fermentation; CHP (7)—cherry pomace addition on the seventh day of fermentation.

### 3.2. Sensory Evaluation

The results in Figure 1 show that the cherry products' addition did not affect the beer aroma intensity or the carbonation, but it led to a decrease in beer body perception, a reduction in beer bitterness, and to the appearance of fruity notes. It has to be noted that the taste of the beer with cherry juice added at the beginning of fermentation (CHJ 1) was the most bitter. The addition of cherry pomace at the beginning of fermentation (CHP 1) resulted in the highest mark for sour taste and aftertaste. However, the beers with cherry pomace (CHP 1 and CHP 2) had a better aroma/taste balance and a more harmonious aroma and taste, compared to those with cherry juice (CHJ 1 and CHJ 7) addition. The sample with pomace added at the beginning of fermentation (CHP 1) stood out from the rest. Its ratings were the closest to the control one, and it was even preferred over the control due to the more pronounced fruity notes (Table 3).

**Table 3.** Ranking sensory test of the beer samples.

| Rank [1] | I | II | III | IV | V |
|---|---|---|---|---|---|
| Sample | CHP (1) | Control | CHP (7) | CHJ (7) | CHJ (1) |

[1] Rank I—the most preferred sample; rank V—the least preferred sample. CHJ (1)—cherry juice addition on the first day of fermentation; CHJ (7)—cherry juice addition on the seventh day of fermentation; CHP (1)—cherry pomace addition on the first day of fermentation; CHP (7)—cherry pomace addition on the seventh day of fermentation.

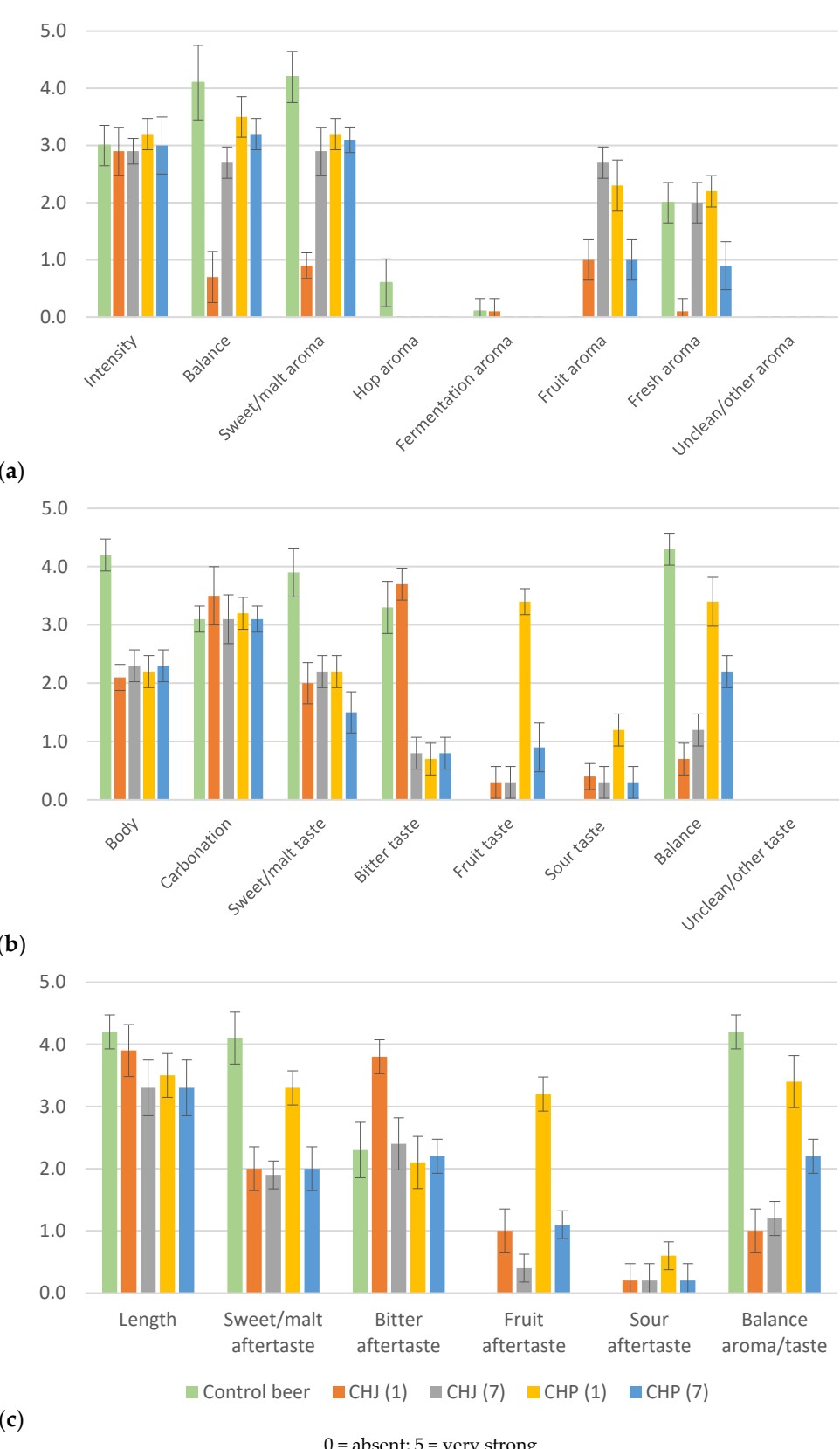

(a)

(b)

(c)

Control beer    CHJ (1)    CHJ (7)    CHP (1)    CHP (7)

0 = absent; 5 = very strong
The absence of data on the histograms means that the value of that parameter is 0.

**Figure 1.** Descriptive sensory test of the beer samples—(**a**) aroma; (**b**) taste; (**c**) aftertaste. CHJ (1)—cherry juice addition on the first day of fermentation; CHJ (7)—cherry juice addition on the seventh day of fermentation; CHP (1)—cherry pomace addition on the first day of fermentation; CHP (7)—cherry pomace addition on the seventh day of fermentation.

### 3.3. Phenolic Compounds and Proteins

The results in Figure 2 show that the cherry products' addition led to an increase in TPC, PA, and F concentrations. In beers with cherry juice addition, the concentration of TPC was between 9 and 11% higher than the control sample. The pomace addition resulted in an increase in TPC between 20 and 25%. This was probably due to the additional extraction of phenolic compounds from the pulp and skins of the cherry pomace during fermentation.

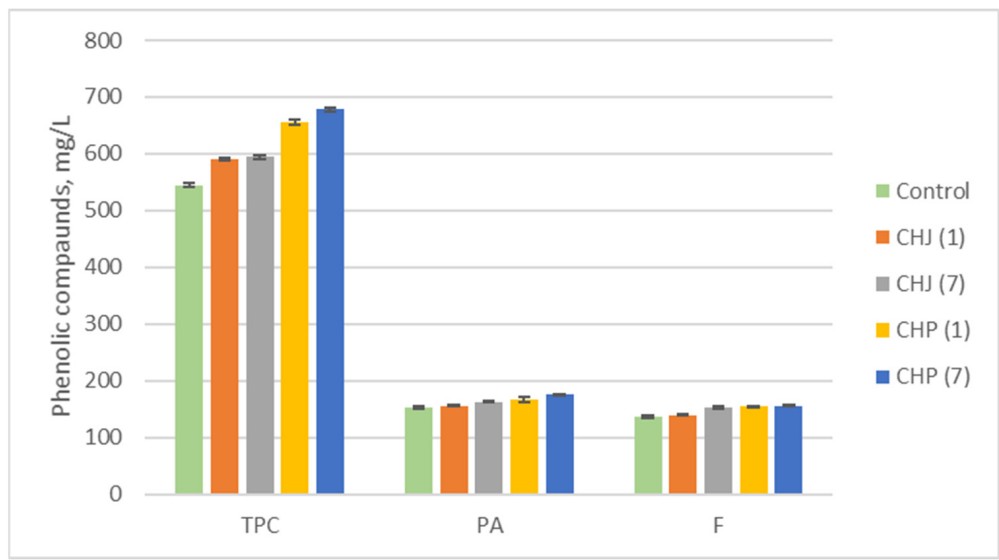

**Figure 2.** Phenolic compounds of the beer samples. TPC—total phenolic compounds; PA—phenolic acid; F—flavonoids; CHJ (1)—cherry juice addition on the first day of fermentation; CHJ (7)—cherry juice addition on the seventh day of fermentation; CHP (1)—cherry pomace addition on the first day of fermentation; CHP (7)—cherry pomace addition on the seventh day of fermentation.

The obtained results for the TPC of the control were comparable to those of Mitič et al. [22] (328–545 mg/L) and were slightly higher than those of Nardini and Garaguso [5] (321–446 mg/L). The data for TPC concentration of beers with cherry products were slightly lower than those obtained by Nardini and Garaguso [5] (747–767 mg/L) and in the range of values obtained by Perez-Alva et al. [8] (35–97 mg/100 mL). The PA concentration in beers with cherry products were within the results obtained by Baigts-Allende et al. [2] for commercial cherry beers (102–319 mg/L). The results for F content in beers with cherry products were lower than commercial cherry beers, obtained by Nardini and Garaguso [5] (196–222 mg/L) and Baigts-Allende et al. [2] (180–636 mg/L). The observed differences can be attributed to the type and quantity of added cherries, the styles of analyzed beers, and the used technologies for beer production.

Regardless of the cherry product type, the increase in phenolic compounds was higher when the fruit product was added on the seventh day of fermentation. Two main hypotheses can be used to explain the results. According to the first one, the higher alcohol content in the medium can help with phenol extraction from the cherry products. According to the second one, the reaction time between phenolic compounds from the cherry products and proteins from the beer was very short. The last hypothesis was confirmed by the data on the protein concentration in the studied beers (Table 4).

It can be seen that the addition of cherry products significantly reduced the protein content of beer. The cherry juice addition resulted in a decrease of between 3 and 4 times in protein concentration but the cherry pomace addition led to a protein concentration reduction of between 8 and 18 times. The addition of the fruit product at the beginning of the fermentation was accompanied by higher losses.

**Table 4.** Protein concentration in the beer samples.

| Beer Sample | Proteins, mg BSA/L |
|---|---|
| Control | $399.5 \pm 6.9$ |
| CHJ (1) | $96.1 \pm 3.6$ |
| CHJ (7) | $148.0 \pm 2.3$ |
| CHP (1) | $22.0 \pm 0.6$ |
| CHP (7) | $48.1 \pm 1.1$ |

CHJ (1)–cherry juice addition on the first day of fermentation; CHJ (7)—cherry juice addition on the seventh day of fermentation; CHP (1)—cherry pomace addition on the first day of fermentation; CHP (7)—cherry pomace addition on the seventh day of fermentation.

The significant decrease in the protein concentration was due to the enrichment of the beers with phenolic compounds from the cherries. It is known that phenolic compounds provoke protein precipitation in beer [23–26]. A decrease in both protein concentrations, the amount and the number of individual protein fractions, in fruit beers was also reported in our previous research [27].

The lower perception of beer body in the samples with cherry product addition, which was reported in Section 3.2, could be ascribed to the significant reduction of proteins in the beers. A similar trend was also observed in our previous research [13,27]. The positive influence of the proteins, especially the medium molecular fraction (15–40 kDa), on the body and head retention of beer was also mentioned by other authors [28–30].

*3.4. Antioxidant Activity*

The AOA of the investigated beers was determined using six different methods. The results showed that the beers with cherry products had a higher AOA than the control sample (Figure 3a,b). The antioxidant activity of beer depends on the antioxidant contents of the used malt and hop and the technological regimes [31]. Cherry products contain a lot of substances that reduce the oxidative stress such as phenolic compounds and anthocyanins. Moreover, the highest concentration of TPC and anthocyanins was reported to be in the skin, followed by the flesh and pits [10]. Therefore, the pomace addition led to a significant increase in the AOA compared to the juice addition. An exception to the stated trend was observed in the results for the AOA measured using the ABTS method (Figure 3a).

The cherry product addition at the later stage of fermentation led to a higher AOA of the samples, measured using the majority of the methods. Again, the results of AOA measured using the ABTS method were an exception. Their tendency was exactly the opposite (Figure 3a).

The AOA of beers was studied by many authors [5,8,22,32]. It was difficult to compare the results obtained because of the differences in beer styles that were analyzed and the used methods for the analysis. Moreover, all of the analyzed beers were commercial. Only the DPPH and ORAC results of our samples were comparable to those obtained by Mitiċ et al. [22] and Perez Alva et al. [8], respectively.

The AOA values of the studied samples increased in the following order: DPPH < ABTS < FRAP < HORAC < CUPRAC < ORAC. Tafulo et al. [33] reported an identical relationship (DPPH < TEAC < FRAP < CUPRAC < TRAP < ORAC) for beers, and Shopska et al. [34] reported an identical relationship (DPPH < ABTS < FRAP < CUPRAC < ORAC) for the AOA of different malt types.

It was interesting to note that there was a very strong correlation between the AOA results obtained using the DPPH, FRAP, CUPRAC, and to some extent, the HORAC method. The correlation coefficients were greater than or equal to 0.85 (Table 5). As it is known, the DPPH, FRAP, CUPRAC, and FC methods are based on a single electron transfer (SET) mechanism, while the ABTS and ORAC methods are based on a hydrogen atom transfer (HAT) mechanism [35]. This may explain the weaker degree of correlation with the last two methods.

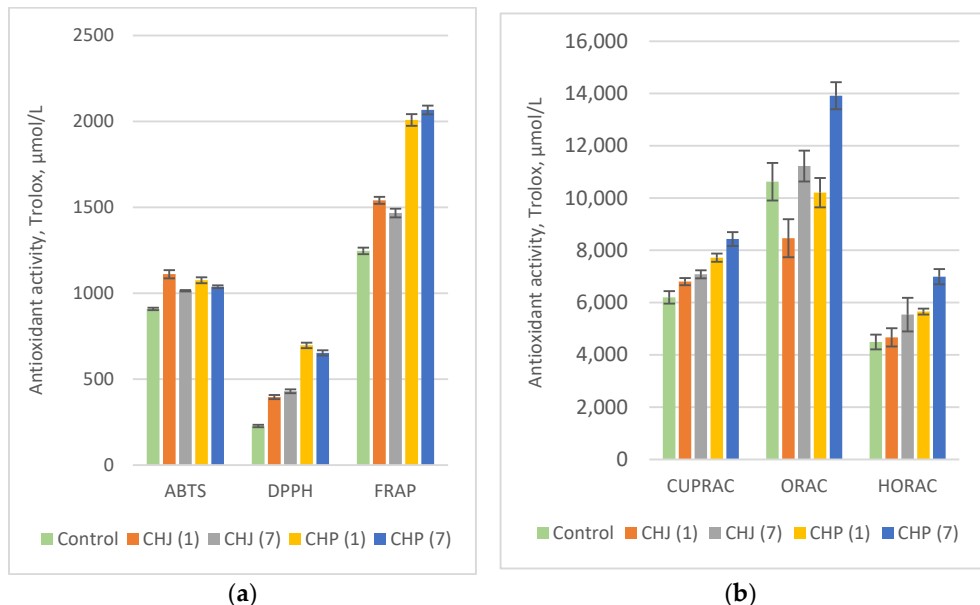

**Figure 3.** Antioxidant activity of the beer samples. (**a**) Antioxidant activity, measured using ABTS, DPPH, and FRAP methods. (**b**) Antioxidant activity, measured using CUPRAC, ORAC, and HORAC methods. CHJ (1)—cherry juice addition on the first day of fermentation; CHJ (7)—cherry juice addition on the seventh day of fermentation; CHP (1)—cherry pomace addition on the first day of fermentation; CHP (7)—cherry pomace addition on the seventh day of fermentation.

**Table 5.** Correlation coefficients between antioxidant activity determined using different methods and phenolic compounds.

|          | ABTS | FRAP | HORAC | CUPRAC | ORAC  | TPC  | PA   |
|----------|------|------|-------|--------|-------|------|------|
| **DPPH**   | 0.60 | 0.98 | 0.80  | 0.93   | 0.41  | 0.97 | 0.89 |
| **ABTS**   | -    | 0.57 | 0.24  | 0.47   | −0.29 | 0.56 | 0.34 |
| **FRAP**   | -    | -    | 0.83  | 0.95   | 0.47  | 0.99 | 0.90 |
| **HORAC**  | -    | -    | -     | 0.95   | 0.85  | 0.89 | 0.99 |
| **CUPRAC** | -    | -    | -     | -      | 0.67  | 0.98 | 0.98 |
| **ORAC**   | -    | -    | -     | -      | -     | 0.55 | 0.77 |
| **F**      | -    | -    | -     | -      | -     | 0.87 | 0.94 |
| **TPC**    | -    | -    | -     | -      | -     | -    | 0.95 |

Furthermore, the data for the AOA correlated very strongly with the concentrations of TPC, PA, and F. Therefore, it could be assumed that the PA and F in the studied beers exhibited their antioxidant activity rather through a SET mechanism and had a good metal-chelating effect. The CUPRAC, FRAP, and HORAC measure the antioxidant capacity based on the metal-chelating activity of antioxidants [35].

## 4. Conclusions

Beers with cherry products (juice and pomace) added on the first and seventh day of fermentation were produced. The addition of cherry juice and pomace led to the appearance of fruity notes and an increase in the phenolic compound content and the AOA of the beer samples. The increase was much more significant and the sensory characteristics were better when the pomace was added. It can be hypothesized that the increase in antioxidant activity after pomace addition will lead to an extension in the beer shelf life. Moreover, cherry pomace is a waste product from cherry juice production, so that made the produced beers sustainable. It can be concluded that the cherry pomace addition at a concentration of 15% (*w/v*) on the seventh day of fermentation resulted in the production of a beer with more phenolic compounds and a higher antioxidant activity. However, additional

experiments will be carried out in order to optimize the amount of added cherry pomace to obtain a beer with a better sensory profile, because the beer with cherry pomace addition on the first day of fermentation was preferred by the tasting panel.

**Author Contributions:** Conceptualization, P.N. and M.K.; methodology, P.N., I.B., V.S. (Vasil Shikov), V.S. (Vesela Shopska) and M.K.; software, M.K.; formal analysis, P.N., I.B., V.S. (Vasil Shikov) and V.S. (Vesela Shopska); investigation, P.N., I.B., M.K., V.S. (Vasil Shikov) and V.S. (Vesela Shopska); resources, M.K.; writing—original draft preparation, P.N. and V.S. (Vesela Shopska); writing—review and editing, M.K.; visualization, P.N.; supervision, M.K.; project administration, M.K.; funding acquisition, M.K. All authors have read and agreed to the published version of the manuscript.

**Funding:** This research was funded by the University of Food Technologies, Plovdiv, Bulgaria (research grant 05/19-H).

**Data Availability Statement:** The data presented in this study are available on request from the corresponding author. The data are not publicly available due to privacy.

**Conflicts of Interest:** The authors declare no conflict of interest.

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
