# Peer review of "Possibilities for Utilization of Cherry Products (Juice and Pomace) in Beer Production"

_beverages, doi:10.3390/beverages9040095_

Round 1

Reviewer 1 Report

Comments and Suggestions for Authors

Dear Authors,

The paper is very interesting and topical.

It concerns the use of cherry pomace, a waste product from cherry juice production, to produce sustainable beers.

Line195: 5 judges are a very low number! Generally, 7 is considered the minimum number of judges for a sensory panel. Please, explain the reasons.

Figures 1-2 and 3: please, change one of the 2 blue color, to improve the understanding of the results.

Fig.1: please, change the type of figure, it is not easy to understand the results.

Table 3: this table is not clear, please improve the description of the results.

Line 220-230: add the abbreviation of the samples in the paragraph 3.2

Table 4: Samples not Sampel 

In the caption of Figure 2 add the meaning of TPC, PA and F.

Figure 3.The results of the ABTS, DPPH and FRAP are not so clear-

Maybe, it is possible to separate these results from the others:  CUPRAC  ORAC HORAC.

Comments on the Quality of English Language

Line 54: The beers produced were compared to a reference beer without cherry products.

Line 211: However not How-ever,

Line 272: has also been reported change in was also reported

Line 275: subsection 3.2., change:  subsection 3.2, 

Line 276: in our previous researches [13, 26]. 

Reviewer 2 Report

Comments and Suggestions for Authors

The manuscript submitted by Shopska et al. mainly deals with “fruit beer with possibilities for utilization of cherry products (juice and pomace) in beer production”.

The paper is well written and logistically organized. However, some problems or errors should be revised according to the following suggestions.

Abstract:

Line11

I think that findings wich support the beneficial effects of industrials or craft beers should be approached with caution: is an alcoholic beverage. Besides there are a lot fruit beers and not all of them have a “refreshing taste”.

Every beer style has is own recipe and taste. The addition of fruits is made in order to obtain different and specific product and not in order to improve taste, shelf life or other characteristics.

Line 13

Maybe there are not a specific bibliography about addiction of cherry pomace in order to obtain fruit beers, but you can take a look at the “Application of white grape pomace in the brewing technology and its impact on the concentration of esters and alcohols, physicochemical parameteres and antioxidative properties of the beer” by Alan Gasinski et al. (Food Chemistry 367 (2022) 130646).

Line 20

The adjunct of fruit rich in polyphenolic compounds in beer clearly increase the final concentration in it. So I suggest to delete the following sentence:

“The phenolic compounds content and AOA increased, when cherries juice or pomace were added”.

Line 23

See line 11

Keywords

As riported by Rose Marie Pangborne in 1989 “…Several occasional users (and misusers) of simply sensory tests for product assessment still refer to their studies with the picturesque but archaic term ‘organolpetic’ to judge with organs. Since sensory receptors, not ‘organs’, responde to temperature, pain, touch, pressure, as well as to chimica stimuli, the more precise adjective ‘sensory’ is recomnded”.

Introduction

Line 27

Fruit beers are a tipical Belgian beer style. In not necessary to speak about Germany beer styles to talk about Belgian beer culture.

Line 36

Fruit beers are not only “craft beers”

Line 39

Beers with cherry are not “a variant of fruit beer”. Is just a fruit beer.

Materials and methods

Line 97

What do you mean with “a part of them were diluited with methanol in a proper ratio” ? Please could you specify the ratio.

Line 113

Space 1 mL

Line 115

Space 7.5 %

Table 2.

Please add a legend to better understand beer samples.

Figures 1, 2 and 3

Please add a legend to better understand beer samples.

Conclusion

See keywords about “organoleptic” word.

Comments on the Quality of English Language

Minor editing of English language required.

Reviewer 3 Report

Comments and Suggestions for Authors

Authors must avoid including abbreviations in the Abstract. One sentence is needed regarding the impact of your findings on the industry. The increased functionality of the product is a good thing but the most important is the technological benefits. Could you state that the increased antioxidant activity and phenolic content might also positively contribute to the shelflife extension of the beer? I think these aspects are the most important for the reader.

Also, I did not get the idea of comparing fruits with cherry pomace. As you used different matrices (juice and pomace), even in the same proportions, you will have different results. Why didn't you choose for the experimental design only different proportions of cherry pomace? Did you eliminate the stones? If yes, please describe the equipment used for this operation.

The pomace obtaining is not entirely described.

Line 62-63 - Give clear information about the chemical parameters of the raw materials (Brix is missing!).

Home Brew 50 - producer, country?

Which was the model and capacity of the stainless steel cylinder capacity? Which sensor was used for the temperature control?

Line 79 - What do you mean by `` smaller stainless steel fermenters``?

Table 1 - the abbreviations are not clear. Please define them in the text and as Table legend (note). Concentration: is it w/w, v/v, etc? Actually, have you tested only two variants? Why? If affirmative, the experiment is not conclusive. 

Explain the meaning of testing so many antioxidant activity methods (six).

Comments on the Quality of English Language

English language improvement is needed.

Round 2

Reviewer 3 Report

Comments and Suggestions for Authors

Unfortunately, the paper was not significantly improved. As I stated before, the experimental design is not constructed properly. You are not able to take the best decision of which variant is the best for beer as you tested only one variant from the same matrix. 

I suggest you to complete (as you also agreed it is necessary) the experimental part with the other variants, improve the discussion and conclusions sections, and resubmit the manuscript. 

Comments on the Quality of English Language

There are still many spelling errors in the manuscript. Please check carefully and improve it.

Round 3

Reviewer 3 Report

Comments and Suggestions for Authors

The perspective was changed according to my observations. Additional experiments are needed in the future. English language still needs to be improved.

Comments on the Quality of English Language

English language still needs to be improved.